# Research and Experiments of Hazelnut Harvesting Machine Based on CFD-DEM Analysis

**Dezhi Ren** [1] , **Haolin Yu** [1], **Ren Zhang** [1], **Jiaqi Li** [2], **Yingbo Zhao** [3], **Fengbo Liu** [2], **Jinhui Zhang** [4] **and Wei Wang** [1,*]

1  College of Engineering, Shenyang Agricultural University, Shenyang 110161, China
2  Liaoning Agricultural College, Yingkou 115009, China
3  Liaoning Jinzhou Panghe Economic Development Zone Management Committee, Jinzhou 121400, China
4  Chaoyang Agricultural Machinery Technology Extension Station, Chaoyang 122000, China
*  Correspondence: wangweisyau@126.com; Tel.: +86-188-0404-5818

**Abstract:** To solve the problem of difficult hazelnut harvesting in mountainous areas of Liaoning, China, a small pneumatic hazelnut harvesting machine was designed, which can realize negative pressure when picking up hazelnut mixtures and positive pressure when cleaning impurities. The key structure and parameters of the harvesting machine were determined by constructing a mechanical model of the whole machine and combining theoretical analysis and operational requirements. To explore the harvesting machine scavenging performance, Liaoning hazelnut No. 3 with a moisture content of 7.47% was used as the experimental object. Firstly, the terminal velocity of hazelnuts and fallen leaves was measured using a material suspension velocity test bench. Secondly, the gas–solid two-phase flow theory was applied comprehensively, and the motion state, particle distribution, and air flow field distribution of hazelnuts from the inlet to the outlet of the pneumatic conveying device were simulated and analyzed using the coupling of computational flow fluid dynamics method (CFD) and discrete element method (DEM) to evaluate the cleaning performance from the perspective of the net fruit rate of hazelnuts in the cleaning box. Finally, a Box–Behnken design experiment was conducted with the sieve plate angle, the distance of the sieve plate, and the air flow velocity as factors and the net fruit rate of hazelnuts as indicators to explore the influence of the three factors on the net fruit rate of hazelnuts. The parameter optimization module of Design-Expert software was used to obtain the optimal combination of parameters for the factors. The experimental results show that the test factors affecting the test index are the following: the air flow rate, the angle of the screen plate, and the distance of the screen plate. The best combination of parameters was an air flow velocity of 14.1 m·s$^{-1}$, a sieve plate angle of 55.7°, and a distance of the sieve plate of 33.2 mm. The net fruit rate of hazelnuts was 95.12%. The clearing performance was stable and can guarantee the requirements of hazelnut harvester operation, which provides a certain theoretical basis for the design of a nut harvester.

**Keywords:** pneumatic; CFD-EDM; simulation analysis; harvester

## 1. Introduction

According to statistics, China has produced 130,000 tons of hazelnuts in recent years, making it the second largest hazelnut producer in the world, with wild mountain hazelnut production accounting for 23% of the total. The picking and sorting of mountain forest hazelnuts is an important part of hazelnut mechanization, and the design of its cleaning device directly affects the working performance of the harvester. Most of the hazelnut harvesters in other countries are adapted from vibratory forest fruit harvesters for flat orchard operations [1–3]. In China, hazelnuts are grown in mountainous areas, so small hazelnut harvesters are suitable for hazelnut harvesting in mountainous area. At present, the hazelnut harvesting operation is divided into two parts, picking and impurity sorting,

and the manual operation faces problems such as high labor intensity and low operational efficiency. Therefore, the mechanization of hazelnut harvesting can solve the above problems.

The agricultural material cleaning process is typically a combined effect of the gas–solid two-phase flow field [4,5]. It is of great significance to study the air-and-screen cleaning device to analyze the air flow distribution of the cleaning shoe and to explore the motion law of agricultural materials on the screen surface, which not only provide the theoretical basis for designing and optimizing the existing typical cleaning unit, but also give theoretical inspiration to look for new cleaning methods. The use of computational fluid dynamics (CFD) for the computation of turbo machinery flows has significantly increased in recent years [6,7]. Furthermore, combined with measurements, CFD provides a complementary tool for simulating, designing, optimizing, and analyzing the flow field inside a turbo machine [8]. The coupling of DEM and CFD provides a means of momentum and energy exchange between solids and fluids, which, in principle, removes the need for some of the semi-empirical approximations employed in CFD solid–fluid models, and is attracting increasing interest from industries. This enables the investigation of fluidized beds, pneumatic conveying, filtration, solid–liquid mixing, and many other systems. Effective modeling of the solid–fluid flow requires methods for adequately characterizing the discrete nature of the solid phase and representing the interaction between solids and fluids. DEM-CFD models reported in the literature have largely been applied to the simulation of fluidized beds and, more recently, to the pneumatic transport of particles [9–16]. Many industrial processes involve complex geometries, often with moving parts, and complex fluid dynamics. For example, the design of pneumatic seed collecting and discharging devices and horizontal seed supply tubes [17–21] coupled with DEM-CFD techniques have also been used as tools to optimize the design of machine structures [22–27].

The air-absorbing hazelnut harvester designed in this paper includes two parts: a conveying pipeline pick-up box and a scavenging box. Because the gas–solid two-phase flow formed during the pneumatic conveying of hazelnuts is more complicated, and at the same time the structure of the scavenging box has a greater impact on the operating effect of this machine, this paper first measured the suspension velocity of typical hazelnuts and fallen leaves by using a suspension test bench to provide data support for the design of the scavenging device. Then, the discrete element method (DEM), computational fluid dynamics method (CFD), and gas–solid two-phase flow theory were used to simulate the flow–solid coupling analysis of the harvester device from the air flow distribution of the cleaning box and the movement law of the material on the screen, in order to obtain the relationship between the movement characteristics of hazelnuts in the conveying process, the flow field distribution characteristics, and the structural parameters of the cleaning box. By analyzing the motion state of the mixture material during the conveying process and the air flow field distribution law of the hazelnut harvester, the effects of the sieve plate angle, the distance of the sieve plate, and the air flow velocity on the net fruit rate of hazelnut were specifically studied, and the potential mechanism was analyzed. The optimal operating parameters of the harvester were determined, and the performance of the harvester was evaluated by prototype tests.

## 2. Materials and Methods

### 2.1. Test Materials and Their Basic Physical Characteristics

Hazelnuts are widely grown in the mountainous area of Liaoning, and in this study, Liaoning hazelnut No. 3 grown in Huanren, Liaoning, was used as the sample. These samples were harvested in September 2022 with a cumulative sample size of 20 kg. The instrument used for the moisture content determination test of hazelnuts and their mixes was the JH-H5 moisture dryer, and the experimental results showed that the water content of hazelnut was 7.47% and the water content of the leaves was 0.53%. The mixture was

classified into two states, hazelnut and leaf, using an electronic balance (BS200S-MEI) and a statistical classification.

The mass and size of the hazelnuts were measured, and according to the measurements it can be concluded that the mass of hazelnuts is between 2 g and 3 g in 70% of fruit. The volumes of hazelnuts were measured, and they can be considered as round; the diameter distribution was between 23 mm and 30 mm in 83% of fruit. Hazelnut harvesters also have leaves mixed in during the harvesting process, as shown in Figure 1.

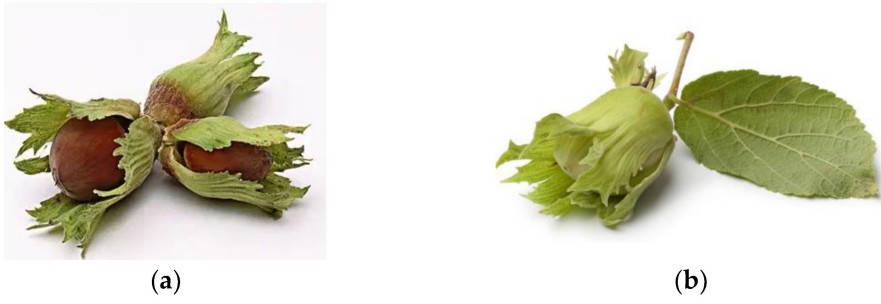

|         (**a**)         |         (**b**)         |

**Figure 1.** Hazelnut, mixed sample chart: (**a**) hazelnut; (**b**) leaves.

### 2.2. Design of the Main Components of the Hazelnut Harvester

The hazelnut harvester achieves separation of the picking device and the wind selection device by using the unloading device, so that the wind of the two devices does not interfere with each other. At the same time, the mixture from the picking device can be transported to the wind sorting device for secondary wind sorting. When the pick-up device picks up, the mixture collides with the screen plate and breaks up the material. The dust-like impurities are sucked out by the negative-pressure air flow of the centrifugal fan, and the rest of the hazelnut mixture will fall into the discharge device and further fall into the wind selection box device for secondary wind selection. The hazelnuts slide down to the hazelnut collection box by gravity, and the remaining leaves and other debris are wind-selected to the miscellaneous collection box by the sieve leaf plate under positive-pressure wind. The angle and position of the sieve plate at each level inside this hazelnut harvester can be freely adjusted to harvest different kinds of nuts. The working principle is shown in Figure 2.

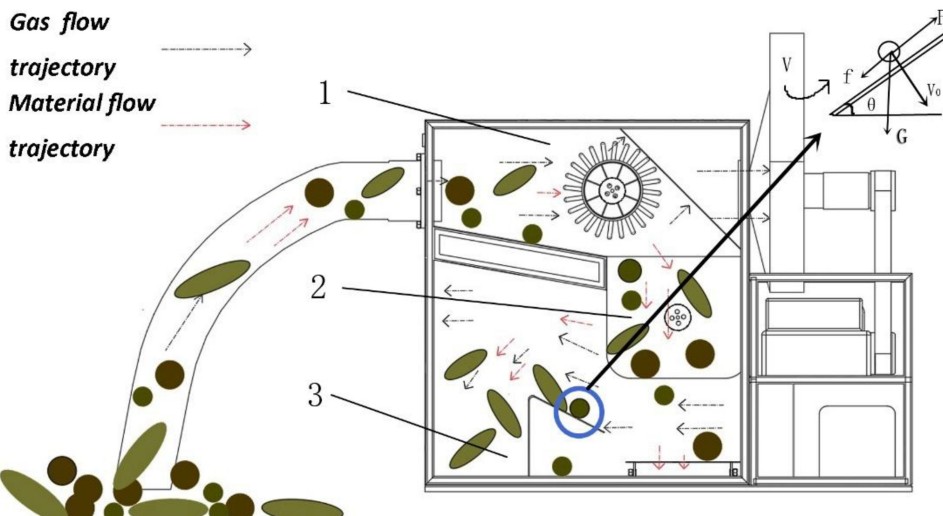

**Figure 2.** Hazelnut harvester gas–material flow trajectory diagram: 1. material picking device, 2. wind sorting device, 3. discharge device.

The designed multi-purpose picking head device includes a harvesting device, transmission device, support device, collection device, and other structural components. The

pick-up device includes eccentric vibration ring, brush head, and other structural components. The collection device includes a fan, collection box, and collection funnel structural components. The transmission device includes a 57 series three-phase hybrid stepper motor, transmission shaft, and other structural components. The support device includes frame, bearing seat, front and rear part of the grip handle, and other structural components. When the hazelnut harvester is working, the handheld multifunctional picking head can pick hazelnuts on the tree, and the brush head part can rotate under eccentric vibration, relying on inertia to remove the hazelnut fruits from the tree. In addition, a small fan on the multi-purpose picking head is designed to provide cooling for the working motor and to help the brush head get rid of twisted branches. This fan can also blow away impurities at any time during the working process and blow down the already combed hazelnut fruit into the collection pipe for the next sorting step of the hazelnut harvester.

The main components of said hazelnut harvester also include structural components such as a picking device, wind selection device, power device, and support device. The picking device includes structural components such as a picking duct, deflector, air regulating valve, fish scale sieve plate (primary sieve plate), and cylindrical roller brush. The wind sorting device includes structural components such as the unloading device, the sieve leaf plate (secondary sieve plate), the hazelnut collection box, and the impurity collection box. The power and support device includes a centrifugal fan, clutch, diesel engine, battery, frame, universal wheel, and other structural components. The hazelnut harvesting machine structure display is shown in Figure 3.

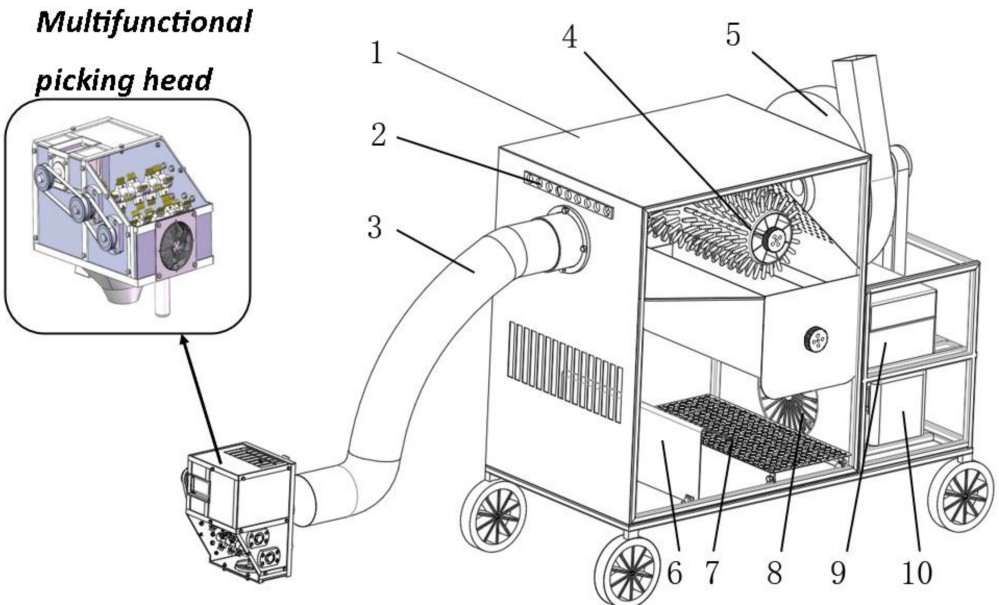

**Figure 3.** Hazelnut harvesting machine structure diagram: 1. machine outer casing, 2. air regulating valve, 3. conveying pipeline, 4. cylinder roller brush, 5. centrifugal fan, 6. sieve leaf plate, 7. vibrating screens, 8. positive-pressure fan, 9. gasoline engine, 10. storage battery.

The discharging device mainly consists of a cavity shell, a fixed blade, and an adjusting blade, as shown in Figure 4. Six cavities of equal volume are formed inside the discharge device, and the regulating blades are made of rubber to prevent damage to the hazelnuts. When working, the particle mixture falls into the discharge device, and when the cavity is opposite to the lower outlet, hazelnuts and fallen leaves are discharged from the discharge device.

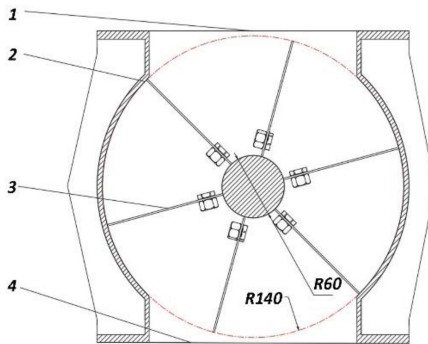 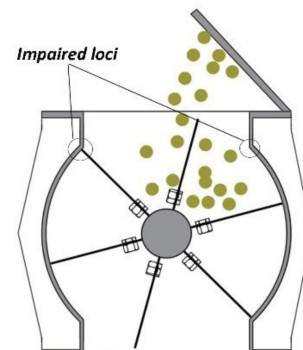

**Figure 4.** Discharge device structure and hazelnut collision schematic: 1. material feed inlet, 2. chamber housing, 3. adjusting blades, 4. material outlet.

The discharging device mainly plays the role of conveying the particle mixture and locking the gas, and the discharging device has an important influence on the working performance of the hazelnut harvesting machine. If discharging is too slow, it can cause hazelnuts and leaves to accumulate in the picking device and reduce conveying efficiency. If discharging is too fast, it is easy to cause the particle mixture to be too late for sorting, and also cause the pressure loss at the connection between the discharge device and the harvester to increase. During the design process, it was found that most of the particle mixture fell on the right side of the cavity shell after hitting the screen plate. When the blade rotates clockwise, the blade, the right damage point, and hazelnuts touch more at the same time. When the blade rotates counterclockwise, the hazelnuts, the left damage point, and the blade touch less at the same time, so the discharging device uses counterclockwise rotation to discharge. The discharging device structure and hazelnut collision are shown in Figure 4.

The discharge volume of the discharge device should meet the requirements of pneumatic conveying volume, which can be calculated by using Formula (1).

$$Gs = 0.06n\Psi \, i\gamma s \tag{1}$$

In the formula, $Gs$ is the discharge volume, kg/h; $n$ is the impeller speed, 25 r/min; and $\Psi$ is the impeller filling factor, 0.6~0.8.

The value of 0.8 for is granular materials; $i$ is the effective discharge volume of the impeller, m³; and $\gamma s$ is the capacity of the conveyed material, 850 kg/m³.

$$i = (R - r)[\pi(R + r) - \xi z \,] \tag{2}$$

In the formula, $R$ is the radius of the outer edge of the impeller, 0.14 m; $r$ is the radius of the root of the impeller, 0.06 m; $\xi$ is the thickness of the blade, 0.01 m; $z$ is the number of blades, 6; and $L$ is the blade length, 0.23 m.

$Gs$ = 640 kg/h is obtained by Equations (1) and (2). Considering that impurities are picked up together with the picking process, the unloading volume should be 1.2~1.5 times the pneumatic conveying volume, and the obtained $Gs$ value meets the design requirements.

The wind sorting device is one of the main structures of the hazelnut harvester. In order to improve the net fruit rate of the wind sorting device, the air flow sorting device with negative-pressure picking and positive-pressure cleaning was designed according to the fluid flow principle. The picking device mainly consists of a fish scale sieve plate and air regulating valve. The fish scale sieve plate is at a 30° angle, and according to the size of the picking chamber design the rectangular fish scale sieve plate length is 600 mm and the width is 250 mm. The aperture of the fish scale sieve is a 15 mm, 15 × 10 arrangement. According to the characteristics of the material through mechanical analysis, it is concluded that the screen leaf plate is at an angle of 55° with the bottom. The aperture of the vibrating screen plate is a 17 mm, 15 × 30 arrangement, mainly to distinguish the size of the hazelnuts.

In order to explore the mechanical mechanism affecting the hazelnut picking and sorting process, the critical velocity of the sucking up and sorting of hazelnuts and fallen leaves is calculated according to the principle of gas–solid two-phase flow, as follows.

$$F_x = \frac{1}{8}\Pi\mu\rho y^2 v_q^2 \tag{3}$$

$$F_z = \Pi g \rho x y^2 \tag{4}$$

$$G = m_{\max}g \tag{5}$$

$$G = F_x + F_z \tag{6}$$

Simplification of Equations (3) and (6) of critical conditions for hazelnut suspension velocity:

$$v_q = \sqrt{\frac{8m_{max}g - 2\Pi g\rho xy^2}{\Pi\mu\rho y^2}} \tag{7}$$

$$v = kv_q \tag{8}$$

In the formula, $F_x$ is the attractive force (N); $F_z$ is the resistance (N); $\mu$ is the resistance constant; $\rho$ is the air density (kg·m$^{-3}$); $x$ is the short axis diameter of the material (mm); $y$ is the long axis diameter of the material (mm); $v_q$ is the flow velocity of the theoretical air flow (m·s$^{-1}$); $v$ is the flow velocity of the actual air flow (m·s$^{-1}$); $m$ is the mass of the material (g); $m_{max}$ is the maximum mass of the material (g); $g$ is the acceleration of gravity (m·s$^{-2}$); and $k$ is the reliability coefficient.

According to the test that measured the average mass of hazelnuts, $m$ is 3.61 g, the average mass of fallen leaves is 2.3 g, the drag coefficient $C$ is 0.6, the projected area of hazelnut in the direction of motion $S$ is 3.1 mm$^2$ and fallen leaves is 12.56 mm$^2$, the critical condition of hazelnut suspension speed that can be calculated from the above formula is 15.6 m·s$^{-1}$, and the critical value of fallen leaves suspension speed is 4.92 m·s$^{-1}$.

To verify the theoretical calculation value, the suspension speed experiment was conducted at the Agricultural Machinery Laboratory of Shenyang Agricultural University. The instruments used were the material suspension speed test bench of model PS-20 and an electronic balance (BS200S-MEI). In order to ensure the accuracy of the test, each group of materials was tested five times repeatedly, three measurement points were selected and averaged for each group of tests, and the final suspension speed of each material was similar to the theoretical calculated value, which can be used as a basis for design.

The hazelnut gravitational component force $g\cos\theta$ can be considered to be balanced with the Magnus effect force $F_M$; in the direction of hazelnut motion by the joint action of the differential pressure force generated by the air flow and $g\sin\theta$, the differential equation for the motion of the hazelnut is obtained according to D'Alembert's principle:

$$\frac{dv_{sx}}{dt} = \frac{F_M}{m} + g\cos\theta \tag{9}$$

$$\frac{dv_{sy}}{dt} = \frac{k\rho S(v - v_s)^2}{m} - g\sin\theta \tag{10}$$

In the equation, $F_M$ is the Magnus effect (N) and $v_s$ is the hazelnut velocity (m·s$^{-1}$).

In the sorting process, hazelnuts will be in contact with the sieve plate, in addition to the above forces, but are also affected by the sieve plate support force $F_N$ and friction resistance $F_f$, according to D'Alembert's principle to obtain the differential equation of hazelnut motion, as follows:

$$F_f = \mu F_N \tag{11}$$

$$\frac{dv_{sx}}{dt} = \frac{F_M + F_N}{m} + g\cos\theta \tag{12}$$

$$\frac{dv_{sy}}{dt} = \frac{k\rho S(v - v_s)^2}{m} - gsin\theta + \frac{\mu F_N}{m} \tag{13}$$

According to Equation (13), the velocity of hazelnut movement in the sorting device is influenced by the average velocity of gas flow $v$, the initial average velocity of hazelnut $v_s$, and the angle $\theta$ between the sieve plate and the horizontal direction. Therefore, the factors affecting hazelnut sorting mainly include gas flow velocity and sieve plate angle, which are controlled by fan speed and sieve plate angle in the experiment. The theoretical analysis of fluid dynamics provides the theoretical basis for the simulation and experimental design.

### 2.3. Theoretical Analysis of CFD-DEM Coupling

Fluent is one of the most functional and applicable CFD software in recent years, and it has good applications mainly in industries related to fluids, heat transfer, and chemical reactions. The discrete element simulation software Rocky is a general CAE software based on the discrete element method, which can be used to simulate and analyze the mechanical behavior of granular materials and their effects on material handling equipment. It has been widely used in agricultural machinery, mining equipment, chemical and food grade pharmaceuticals, and other fields Many processes in various industries involve the simultaneous flow of fluids and particles. In these cases, it is important to consider the fluid flow in order to first obtain the correct particle behavior. This is then determined by the particle-level interactions between the fluid, the particles, and the boundaries. Therefore, it is obvious that a modeling approach is needed to deal with particle fluid systems, and there are two methods commonly used to solve them: Eulerian and Lagrangian methods.

In the Eulerian approach, both the fluids and solid phases are treated as interpenetrating continua in a computational cell that is much larger than the individual particles, but still small compared to the size of the process scale. Therefore, continuum equations are solved for both phases with an appropriate interaction term to model them. This in turn means that constitutive equations for inter- and intraphase interactions are needed. Since the volume of a phase cannot be occupied by the other phases, the concept of phasic volume fraction is introduced. Location-based mapping techniques are applied, and local mean variables are used in order to obtain conservation equations for each phase. The advantage of this approach is its reasonable computational cost for practical application problems, making it the most used granular-fluid modeling technique in use today.

In the Lagrangian approach, the fluid is still treated as a continuum by solving the Navier–Stokes equations, while the dispersed phase is solved by tracking a large number of particles through the flow field. Each particle (or group of particles) is individually tracked along the fluid phase by the result of forces acting on them by numerically integrating Newton's equations that govern the translation and rotation of the particles. This approach is made considerably simpler when particle–particle interactions can be ignored. This requires that the dispersed second phase occupies a low volume fraction, which is not the reality in the majority of the industrial applications. Due to the fact that no particle interaction is resolved, the model is inappropriate for modeling applications where the volume fraction of the second phase cannot be ignored, such as fluidized beds. For applications such as these, particle–particle interactions need to be taken into account when solving the dispersed phase. Now, numerous authors have published their work using the Euler–Lagrange type of model to study granular flow [28–31].

The coupled CFD-DEM approach is an effective alternative for modeling particulate fluid systems because it captures the discrete nature of the particle phase while maintaining computational tractability. This is achieved by solving the fluid flow at the cell level rather than at the detailed particle level. By reducing the required fluid calculations, this technique expands the range of devices and processes that can be studied with numerical simulations. In the coupled CFD-DEM approach, the fluid flow is obtained by the traditional continuous medium approach, providing information to calculate the fluid forces acting on individual particles, while the particle motion is obtained by using the discrete particle approach. The gas phase is numerically simulated by CFD, the particles are solved by the DEM

method, and the exchange of energy is coupled through the gas–solid phase interaction. Currently, Rocky has two methods to perform the coupling between particles and fluid: the multiphase coupling approach relies on the Eulerian Multiphase Model of Fluent, where the material particles are represented by another dedicated phase, and the multiphase approach supports an arbitrary number of fluid phases; the single coupling approach is achieved by setting the fluid domain in Fluent as a porous medium, which enables the material particles to influence.

When the Multiphase Model is set to Eulerian in the Fluent case, the averaged mass conservation equation is given by

$$\frac{\partial}{\partial t}\left(\alpha_f \rho_f\right) + \nabla \cdot \left(\alpha_f \rho_f u\right) = 0 \tag{14}$$

whereas the averaged momentum conservation equation is written as

$$\frac{\partial}{\partial t}\left(\alpha_f \rho_f u\right) + \left(\alpha_f \rho_f uu\right) = -\alpha_f \nabla p + \nabla \cdot \left(\alpha_f \Pi_f\right) + \alpha_f \rho_f g + F_{p \to f} \tag{15}$$

where $\alpha_f$ stands for the fluid volume fraction, $p$ is the shared pressure, $\rho_f$ is the fluid density, $u$ is the fluid phase velocity vector, and $T_f$ is the stress tensor of the fluid phase, defined as

$$\Pi_f = \mu_f\left(\nabla u + \nabla u^T\right) + \left(\lambda_f - \frac{2}{3}\mu_f\right)\nabla \cdot u\Pi \tag{16}$$

In Equation (15), $F_{p \to f}$ represents the source term of momentum from an interaction with the particulate phase, calculated according to the expression

$$F_{p \to f} = -\frac{\sum_{p=1}^{N} F_{f \to p}}{V_c} \tag{17}$$

where $V_c$ is the computational cell volume, $N$ is the number of particles inside the computational cell volume, and $F_{f \to p}$ accounts for the forces generated by the fluid on the particles.

When the Multiphase Model is turned off in the Fluent case, Rocky adapts the Fluent setup to treat the DEM particles as a porous media and to assign to the fluid phase momentum and energy source terms (that account for fluid–particle interactions) calculated by Rocky during coupled simulations. The porosity distribution of the domain is a function of the concentration of the solid phase as the simulation progresses.

Considering a single-phase flow through a porous medium and assuming that there is no mass transfer between phases, the averaged mass conservation equation of the fluid phase is given by

$$\frac{\partial}{\partial t}\left(\gamma \rho_f\right) + \nabla \cdot \left(\gamma \rho_f u\right) = 0 \tag{18}$$

where $\gamma$ is the porosity of the medium. Likewise, the averaged momentum conservation equation is

$$\frac{\partial}{\partial t}\left(\gamma \rho_f u\right) + \nabla \cdot \left(\gamma \rho_f uu\right) = -\gamma \nabla p + \nabla \cdot \left(\gamma \Pi_f\right) + \gamma \rho_f g + F_{p \to f} \tag{19}$$

and the averaged energy conservation equation is

$$\frac{\partial}{\partial t}\left(\gamma \rho_f h_f\right) + \nabla \cdot \left(\gamma \rho_f u h_f\right) = \gamma \frac{\partial p}{\partial t} + \gamma \Pi_f : \nabla u - \nabla \cdot \gamma q_f + Q_{p \to f} \tag{20}$$

The porosity $\gamma$ is defined as the relative volume occupied by the void spaces of the porous region. As a single-phase coupled simulation runs, Rocky estimates the porosity of each cell as

$$\gamma = 1 - \alpha_s \tag{21}$$

where $\alpha_s$ is the local volume fraction of the solid phase at the current time step.

### 2.4. Hazelnut Harvester Simulation Parameters' Setting

The establishment of a discrete element model is mainly divided into two kinds, which are geometric model establishment and particle model establishment. According to the design parameters, the 3D modeling software SolidWorks was used to build the 3D model of the hazelnut harvester and to simplify its model parts. The simplified 3D model was imported into Rocky, and a fluid domain model was created. The fluid domain model was meshed with a structured meshing method, with a total of 2.57 million mesh cells and 460,000 nodes. The fluid domain model is shown in Figure 5.

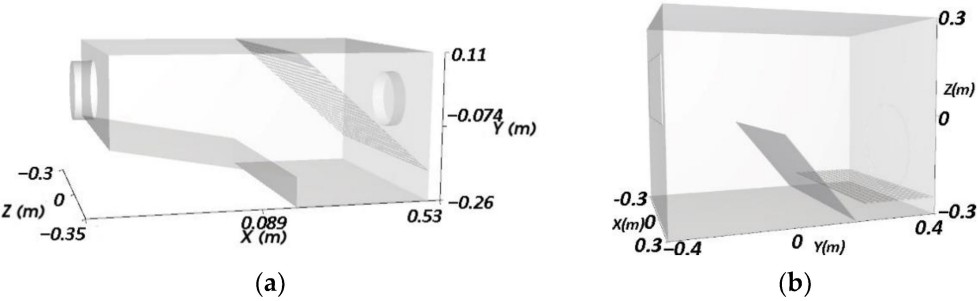

(a)                                    (b)

**Figure 5.** Harvester fluid domain model: (**a**) pickup device fluid domain model; (**b**) fluid domain model of wind separator.

Based on Rocky software, a three-factor, three-level Box–Behnken simulation experiment was conducted with the sieve plate angle, distance of the sieve plate, and air flow rate as factors and hazelnut net fruit rate as the index. The simulation experimental results provide guidance and comparative verification for subsequent field tests. The establishment of the material particle model in Rocky is the basis of the simulation analysis, and the realistic degree of its model directly affects the simulation results. In this paper, we take Liaoning hazelnut No. 3 as the research sample, through the dimensional analysis of hazelnut and leaf drop, where the average diameter of hazelnuts is 22 mm as a round ball, and the average diameter of leaf drop is 36 mm as a disc shape with a thickness of 0.5 mm. In Rocky, the stem model of particles can be created directly, and the shape of leaf drop is sphero-polygon, with a vertical aspect ratio of 1.8, a horizontal aspect ratio of 0.1, and an angle number of 80; the shape of the hazelnuts is spherical. The particle models are shown in Figure 6. In order to distinguish the size of hazelnuts, their diameter was set at 80% of hazelnuts above 18 mm and the rest at 20%.

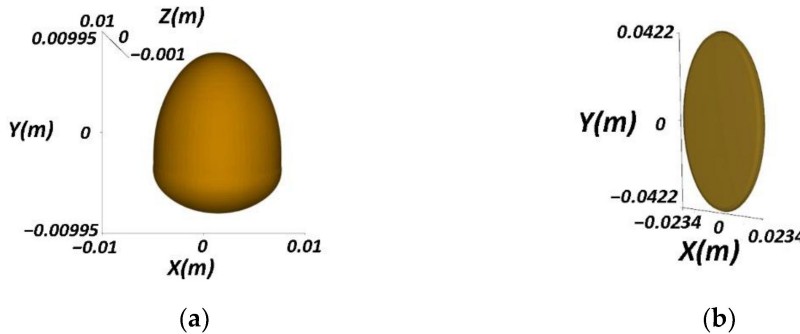

(a)                                    (b)

**Figure 6.** Particle mixture model: (**a**) hazelnut model; (**b**) leaf model.

The material inlet was used to set the simulation particle entry position, direction, and time. A particle inlet in the Geometries module was created, and the entry port was set as a rectangular inlet according to the actual geometry of the feeding device. The length was set as 0.15 m, width as 0.25 m, alignment angle as 90°, and incline angle as 270°. The inlet at the feeding port of the picking device and the feeding port of the sorting device was

set. According to the simulation needs, the feeding time was set to 0~2.5 s. The number of particles was estimated according to the actual machine output, the hazelnuts and leaves in the mixture picked up by the machine were sieved, and the feeding volume was set to 1.6 kg/s for hazelnuts and 0.15 kg/s for leaves. Particle entry simulation parameters are shown in Table 1.

**Table 1.** Particle inlet settings.

| Project Name | Project Setting |
|---|---|
| Entry Point | Inlet |
| Stop Time | 2.5 s |
| Mass Flow Rate (Hazelnut) | 1.6 kg/s |
| Mass Flow Rate (Leaves) | 0.15 kg/s |

In order to ensure the authenticity of the simulation test and reduce the simulation time, hazelnuts and leaves were selected as the study objects in this simulation. The simulation test only considers the interaction between hazelnut, leaf, and hazelnut harvester, ignoring the influence of other impurities on the simulation. Each part of the hazelnut harvester is endowed with material characteristics, and the entire device is made of structural steel. The values of material density, Poisson's ratio, Young's modulus, and dynamic and static friction factors of the particles directly affect the simulation results. We imported the data from the hazelnut stacking angle experiment and the static friction experiment into the EDEM database for analysis and obtained these simulation parameters. The parameters of material mechanical properties and interparticle contact parameters of hazelnuts and leaves are shown in Table 2.

**Table 2.** Simulation parameters.

| Type | Density/(g·cm$^{-3}$) | Poisson's Ratio | Young's Modulus/Pa |
|---|---|---|---|
| Hazelnut | 5.5 | 0.41 | $3.1 \times 10^9$ |
| Leaf | 0.7 | 0.38 | $2.71 \times 10^5$ |
| **Type** | **Dynamic Friction Coefficient** | | **Static Friction coefficient** |
| Hazel–Hazel | 0.06 | | 0.45 |
| Hazel–Leaf | 0.06 | | 0.4 |
| Leaf–Leaf | 0.05 | | 0.42 |

*2.5. Box–Behnken Experimental Design*

The experiment was conducted in October 2022 in a hazelnut orchard in the Heishan region of Liaoning Province, China, as shown in Figure 7.

The experiment was designed according to GB/T5667-2008 "Agricultural Machinery Production Test Methods" and with reference to the GB/T5262-2008 "General Provisions for the Determination of Test Conditions of Agricultural Machinery" standard. Each group of tests collected 1 m$^2$ of fallen hazelnut surface, and each group was repeated three times, setting the picking mouth 20 mm above the ground, taking the average value as the test results recorded and analyzed, with the net fruit rate of hazelnut as the evaluation index. After the first harvest the overall weight in the hazelnut collection box was weighed as $M_{total}$, the weight of hazelnuts was manually selected as $M_{Hazel}$. Then, the total mass of the hazelnut collection box and the miscellaneous collection box in the hazelnut picking and sorting machine was weighed as $M_{pick}$, the surface of the plot was harvested manually again using manual harvesting, and the mass of the manual harvesting was $M_{miss}$. The hazelnut net fruit rate $Y$ was calculated by the following equation.

$$Y = M_{Hazel}/M_{total} \times 100\%$$

The factors influencing the hazelnut harvester were selected and combined with the previous CFD-DEM simulation analysis of the sieve plate angle, the distance of the sieve plate, and the air flow velocity factors. The sieve plate angle was set at 51–59°, the distance of the sieve plate was 17–46 mm, and the air flow velocity was 9–18 m·s$^{-1}$, and a three-factor, three-level Box–Behnken test was conducted. The experimental factor coding table is shown in Table 3.

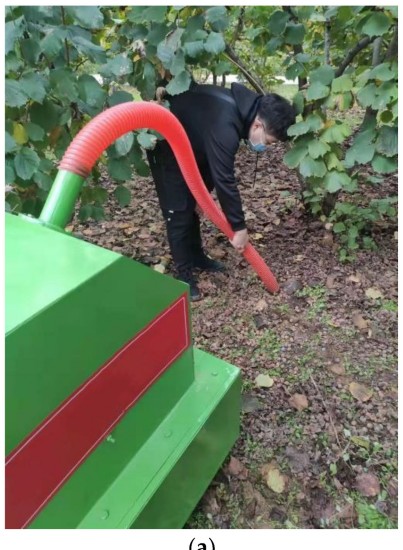
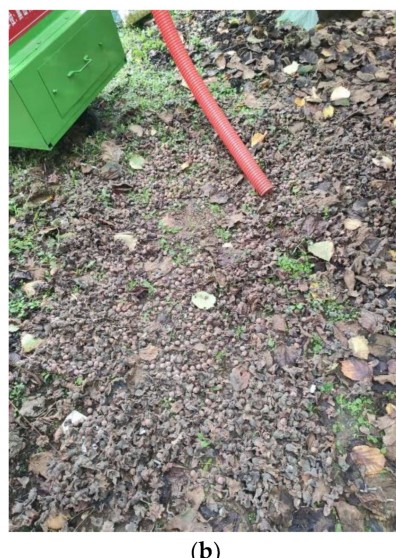

| (a) | (b) |

**Figure 7.** Machine field trials: (**a**) hazelnut picking and sorting test; (**b**) harvester operating site.

**Table 3.** Experimental factor and level table.

| Level | Factor | | |
| --- | --- | --- | --- |
| | Sieve Plate Angle $A/(°)$ | Distance between Sieve Plates $B$/mm | Air Flow Velocity $C/(m·s^{-1})$ |
| −1 | 51 | 17 | 9 |
| 0 | 55 | 31.5 | 13.5 |
| 1 | 59 | 46 | 18 |

## 3. Results and Discussion

### 3.1. Basic Physical Characteristics of the Experimental Samples

The basic physical parameters of the particles were derived from the statistics and measurements of the harvested samples, as shown in Table 4.

**Table 4.** Measurement of physical parameters of hazelnut harvesting material.

| Type | Quantity/ Each | Average Length/cm | Average Diameter/cm | Average Mass/g | Density/ $(g·cm^3)$ |
| --- | --- | --- | --- | --- | --- |
| Simple fruit | 1894 | 1.83 | 1.73 | 2.8 | 1.3 |
| Leaf | 521 | 4.5 | 3.6 | 1.07 | 0.7 |

### 3.2. Analysis of Simulation Results

#### 3.2.1. Terminal Speed Analysis of the Pick-Up Device

In Fluent, the SIMPLE algorithm is used, the mesh model is a hexahedral mesh, and the simulation model is a standard k-epsilon model. The inlet is set as the velocity inlet, the velocity is consistent with the initial velocity of the Rocky particle inlet, the velocity is 20 m·s$^{-1}$, and the fan port is set as the pressure outlet condition. In Rocky, the particle inlet initial velocity is set to be the same as the inlet wind velocity in Fluent, in the direction normal to the inlet plane. Air flow velocity clouds were obtained for the hazelnut harvester

at three air flow speeds, as shown in Figure 8. In the simulation experiment, there was no adhesion between the hazelnut and the leaf, and a contact model without sliding was used between the particles and the wall, with the gravitational acceleration direction along the negative direction of the Z-axis. The feeding time was 2.5 s, total feeding 1.7 kg, total simulation time 3 s, and interval time 0.05 s.

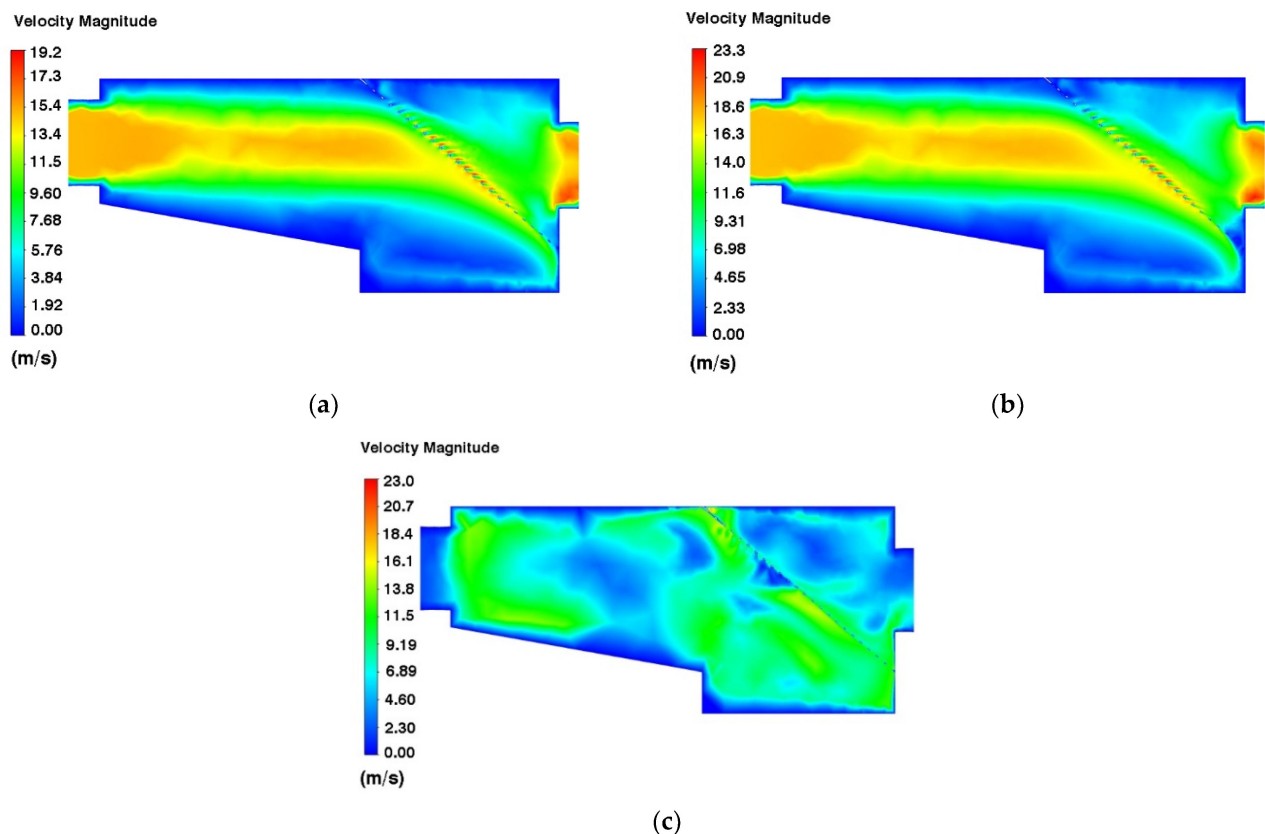

(**a**)                    (**b**)

(**c**)

**Figure 8.** Velocity contour map of the pick-up device: (**a**) air velocity 15; (**b**) air velocity 18; (**c**) air velocity 21.

### 3.2.2. Analysis of the Working Process of the Sorting Device

The particle flow trajectory inside the hazelnut harvester corresponding to 0~3 s time was obtained by coupling simulation with Fluent fluid dynamics software and Rocky discrete element software. The working process is shown in Figure 9. At t = 0.55 s, the particles began to fall by the discharge device, and the particle mixture was blown to the sieve plate by the positive-pressure wind. After the leaves and hazelnuts hit the sieve plate, the lighter leaves were blown to the miscellaneous collection box in front, the gravity of the hazelnuts themselves was greater than the blowing force of the air flow, and the hazelnuts fell by gravity to the hazelnut collection box. At t = 1.65 s, the particle mixture was further separated through the sieve plate. The lighter leaves were partly blown to the debris collection box in front and partly blown out through the air outlet, and more hazelnuts fell by gravity to the vibrating screen waiting for further sorting. At t = 2.55 s, the number of falling particles reached 1.7 kg. The hazelnut collection box had a small number of leaves, and the vibrating sieve further separated the hazelnuts by size. The device in line with the expected design effect.

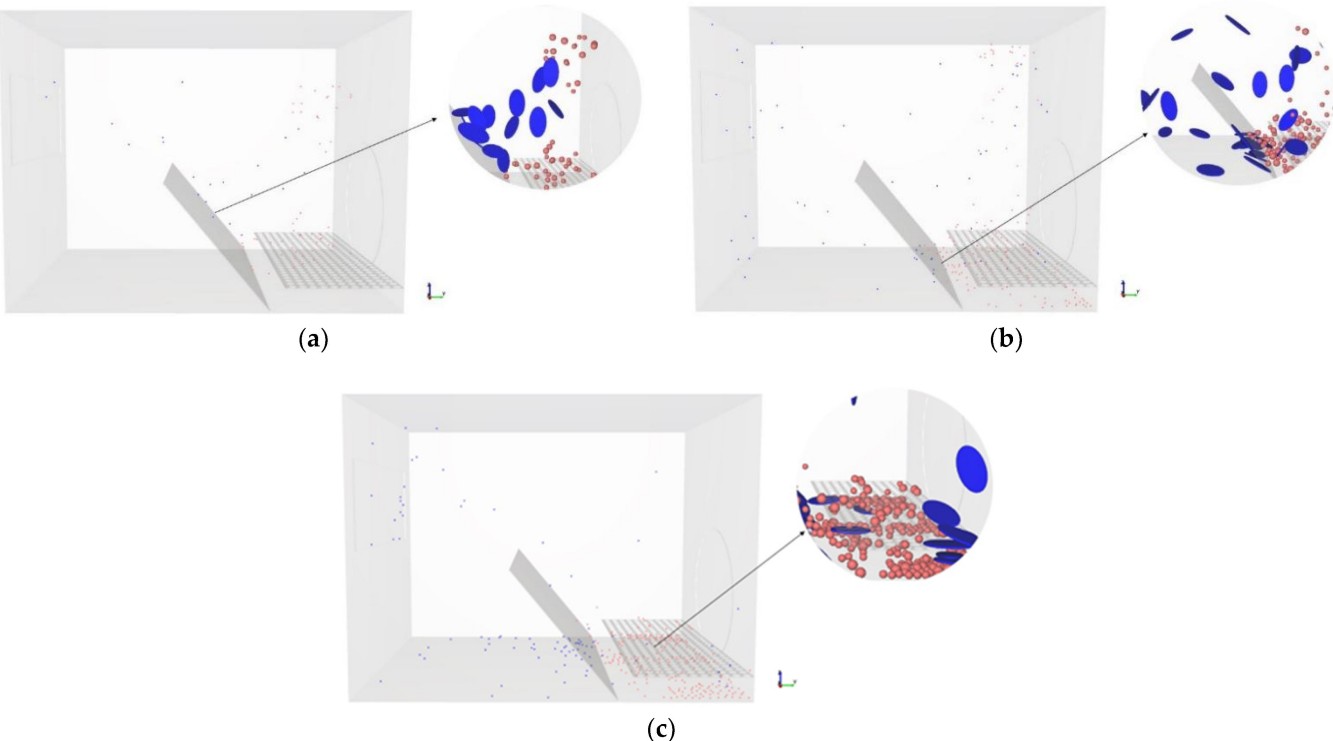

(a)

(b)

(c)

**Figure 9.** The working process of wind sorting device: (**a**) start sorting at t = 0.55 s; (**b**) during sorting at t = 1.65 s; (**c**) end sorting at t = 2.55 s.

Simulation of the whole process was completed for a total of 3 s with a sieve leaf plate angle of 50~60 °, air flow velocity of 10~20 m·s$^{-1}$, and distance of the sieve plate of 15~50 mm for seventeen groups of simulation. The results show that the particles with different densities can be sorted by the screening of the baffles. The comparison curves of the total mass of material with time between the simulation experiment of the sorting device and the field experiment are shown in Figure 10. The trend of the total mass of the particle mixture in the pick-up device with time is shown in Figure 11. The results show that the sorting effect is most obvious, and the net fruit rate is high at the sieve plate angle of 55°, air flow velocity of 15 m·s$^{-1}$, and distance of the sieve plate of 30 mm.

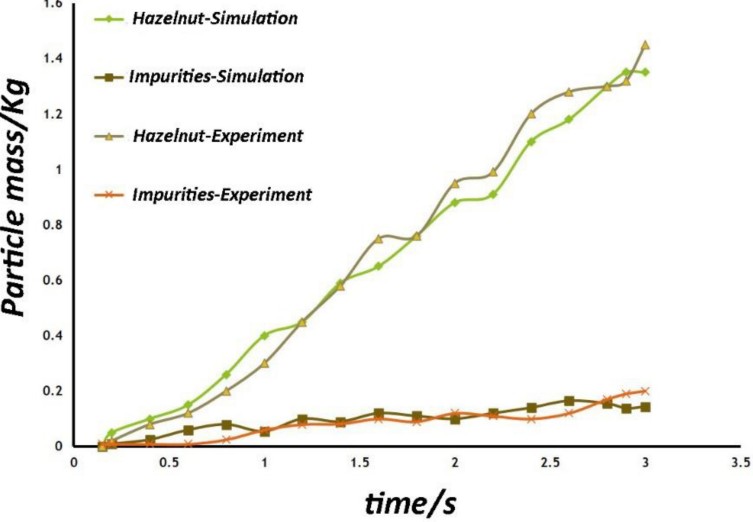

**Figure 10.** Variation in the mass profile of each material in the sorting device.

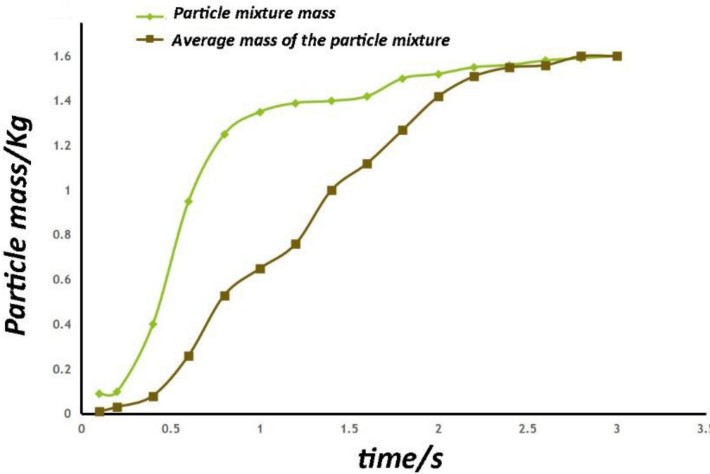

**Figure 11.** Variation in the total material mass profile in the pick-up device.

The coupled CFD-DEM simulation analysis shows that after the granular mixture material goes down from the discharge device, the mixture material collides with the screen plate, the leaves are blown to the impurity collection box with the air flow, and the hazelnuts flow to the hazelnut collection box under the action of gravity. The air flow in the designed sorting device model follows the expected trajectory and the particle mixture is effectively separated. The CFD-DEM simulation experiment obtained the distribution and movement of particles in the separation device, and the overall process of simulation is reasonable and more consistent with the actual working conditions, which can be used to simulate the harvesting and sorting process of hazelnuts.

### 3.3. Analysis of Experimental Design Results

According to the data samples in Table 5, the quadratic polynomial regression model of net fruit rate with three factors was obtained using the data processing software Design-Expert 8.0, and the regression equation was the following:

$$Y = 94.76 + 1.64A + 1.11B + 2.35C - 1.13AB - 1.16AC + 1.55BC - 3.69A^2 - 4.79B^2 - 8.87C^2$$

**Table 5.** Experiment design and result.

| Test Number | Factor | | | Test Result |
|---|---|---|---|---|
| | **A** | **B** | **C** | **Net Fruit Rate Y** |
| 1 | 51 | 31.5 | 9 | 76.9% |
| 2 | 59 | 31.5 | 18 | 84.3% |
| 3 | 55 | 46 | 18 | 86.2% |
| 4 | 55 | 31.5 | 13.5 | 93.3% |
| 5 | 55 | 31.5 | 13.5 | 95.3% |
| 6 | 55 | 17 | 18 | 81.6% |
| 7 | 55 | 17 | 9 | 79.1% |
| 8 | 55 | 31.5 | 13.5 | 95.5% |
| 9 | 51 | 31.5 | 18 | 83.9% |
| 10 | 51 | 46 | 13.5 | 87.4% |
| 11 | 59 | 46 | 13.5 | 88.1% |
| 12 | 55 | 46 | 9 | 77.5% |
| 13 | 59 | 17 | 13.5 | 87.4% |
| 14 | 59 | 31.5 | 9 | 83.7% |
| 15 | 51 | 17 | 13.5 | 82.2% |
| 16 | 55 | 31.5 | 13.5 | 95.3% |
| 17 | 55 | 31.5 | 13.5 | 94.4% |

In the formula, Y is the net fruit rate of hazelnuts; A is the sieve plate angle; B is the distance of the sieve plate; and C is the air flow velocity.

The regression model ANOVA and significance test results are shown in Table 6. From Table 6, it is clear that the fit of the net fruit rate model is highly significant ($p < 0.01$). The regression equation misfit was not significant, and it was a good fit with the actual situation. The *p*-values of the sieve plate angle, distance of the sieve plate, and air flow velocity could determine the effects of the three test factors on the net fruit rate of hazelnuts. The *p*-value of regression term B was less than 0.05 and the effect was significant, the *p*-values of regression terms A and C were less than 0.01 and the effects were highly significant, and the effects of the other terms were significant or highly significant. The effects of the test factors on the net fruit rate were air flow velocity, sieve plate angle, and distance between sieve plates in descending order.

**Table 6.** Analysis of variance.

| Source | Net Fruit Rate | | | |
|---|---|---|---|---|
| | SS | df | F | *p*-Value * |
| Model | 632.23 | 9 | 78.37 | <0.0001 |
| A | 21.45 | 1 | 23.93 | 0.0018 |
| B | 9.90 | 1 | 11.05 | 0.0127 |
| C | 44.18 | 1 | 49.29 | 0.0002 |
| AB | 5.06 | 1 | 5.65 | 0.0491 |
| AC | 10.24 | 1 | 11.42 | 0.0118 |
| BC | 9.61 | 1 | 10.72 | 0.0136 |
| $A^2$ | 57.41 | 1 | 64.05 | <0.0001 |
| $B^2$ | 96.71 | 1 | 107.89 | <0.0001 |
| $C^2$ | 331.08 | 1 | 369.37 | <0.0001 |
| Residual | 6.27 | 7 | | |
| Lack of fit | 2.89 | 3 | 1.13 | 0.4361 |
| Error | 3.39 | 4 | | |
| Sum | 638.51 | 16 | | |

* $p < 0.01$ means extremely significant, $0.01 < p < 0.05$ means significant, $p > 0.05$ means not significant, SS means Sum of Squares, df means Degree of Freedom, F means F-value

Using Design-Expert 8.0 software to process the data and analyze the relationship between the test index and factors, the effect of sieve plate angle, air flow velocity, and distance of the sieve plate on the net fruit rate can be obtained, and the response surface is shown in Figure 12, fixing the factors of sieve plate angle, air flow velocity, and distance of the sieve plate as the 0 level, respectively, and analyzing the interaction between the remaining two factors on the net fruit rate according to the response surface plot. The effect of the interaction between the remaining two factors on the net fruit rate was analyzed according to the response surface.

The response surface of the interaction between air flow velocity and distance of the sieve plate on the net fruit rate is shown in Figure 12a. When the air flow rate is certain, the net fruit rises first and then falls with the increase in distance of the sieve plate; when the distance of the sieve plate is certain, the net fruit rises first and then falls with the increase in air flow velocity. The response surface of the interaction effect of sieve plate angle and distance of the sieve plate on the net fruit rate is shown in Figure 12b. When the sieve plate angle is certain, with the increase in distance of the sieve plate, the net fruit rises first and then falls; when the distance of the sieve plate is certain, with the increase in sieve plate angle, the net fruit rises first and then falls. The response surface of the interaction between air flow velocity and sieve plate angle on the net fruit rate is shown in Figure 12c. When the air flow velocity is certain, with the increase in sieve plate angle, the net fruit rises first and then decreases. When the sieve plate angle is certain, the net fruit rate also rises first and then decreases with the increase in air flow velocity. Based on the interaction effect analysis, it can be seen that when any of the factors of air flow velocity, sieve plate angle,

and distance of the sieve plate are fixed, the interaction of the remaining two factors is first increased to a certain value so that the net fruit effect in the impurity sorting process is significant, and then the net fruit rate decreases as the two factors continue to increase.

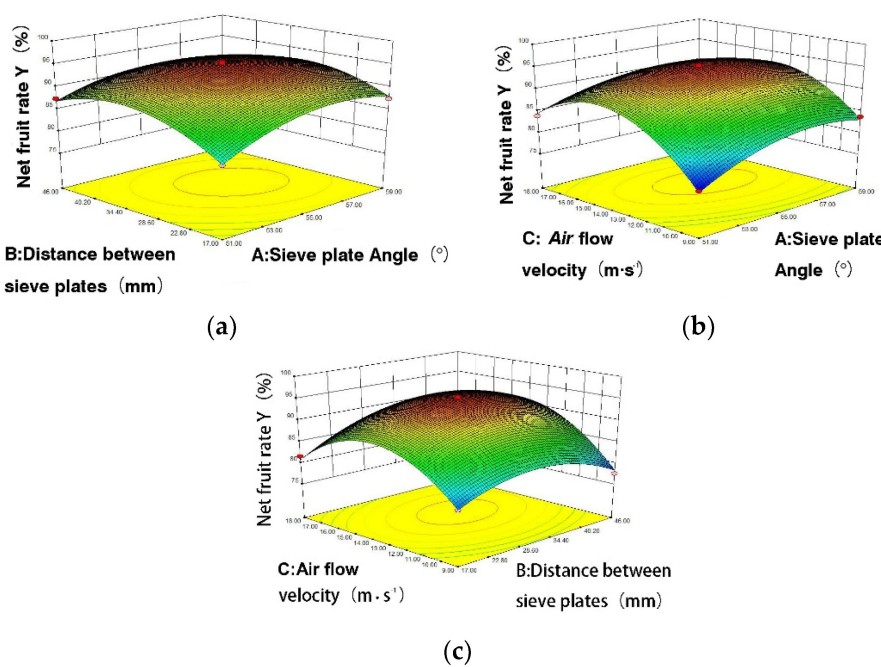

**Figure 12.** Response surface curve of interaction factors for test indexes: (**a**) the effect of AB interaction on Y; (**b**) the effect of AC interaction on Y; (**c**) the effect of BC interaction on Y.

*3.4. Parameter Optimization and Comparison Experiments*

In order to obtain the optimal parameters for each factor, the established three-factor orthogonal test was optimized by using Design-Expert 8.0 software. In order to obtain the optimal parameters for each factor, a three-factor orthogonal test was established using Design-Expert numerical analysis software for parameter optimization. Based on the analysis of the test results and the optimization of the influencing factors, the optimal combination of factors was obtained: the angle of the sieve plate angle was 55.7°, the air flow velocity was 14.1 m·s$^{-1}$, and the distance of the sieve plate was 33.2 mm, which resulted in a prediction of 95.12% of the net fruit yield of hazelnuts.

Based on the optimal parameters obtained, five replicate tests were performed, and the test result was 94.25%. The error rate was 3.4%. It can be seen that under the optimal parameters, the field prototype test results are similar to the simulation test results and can meet the requirements of hazelnut picking operation.

## 4. Discussion

Hazelnuts and leaves were selected from the particle mixture for the study through statistics and analysis of the mixture. The critical value of particle suspension velocity was obtained by the experiment, and it is an important parameter to measure to know whether the material could be separated. A coupled CFD-DEM method was used to simulate the effect of a pneumatic hazelnut harvester on the net fruit rate of hazelnuts under different combinations of operating parameters. The coupled CFD-DEM simulation was verified to describe the motion of the particle mixture in the sorting device well through a field experiment. Through analyzing the fluid dynamics inside the hazelnut harvester, the study proved that the optimization of air flow velocity, sieve plate angle, and distance of the sieve plate could improve the working performance of the hazelnut harvester. The working principle of the hazelnut harvester was analyzed from the point of the movement of the sorting mixture. The results of the comparison experiment show that the hazelnut harvester

with the optimized combination of operating parameters can effectively sort the particle mixture and reduce the impurity rate and energy consumption. Working performance experiments show that the optimized hazelnut harvester could pick and sort the particle mixture well under real field conditions.

In the follow-up research, the combination of various devices should be considered in order to further improve the net fruit rate. For example, HASATSAN researched and designed a high horsepower tractor-driven hazelnut harvester and used a precision-improving interchangeable vibrating screen set to work simultaneously with three harvesting hoses. The harvester was constructed to harvest a particle mixture in any geographic condition with a vacuuming system; TURBO-VAC has designed an integrated hazelnut harvester that used a diesel engine to drive a fan that rotates at high speed to pick and sort particles in a single pass. By refining the functions of each component and the components combining and cooperating with each other, we will be able to further improve the performance of the hazelnut harvester. In this paper only one hazelnut variety was used as the experimental material, and the actual suspension speed of different varieties of hazelnuts varied greatly; hence, further research is needed in terms of applicability. In addition, the vibration of the whole machine and the environment of the actual workplace will have some influence on the cleaning situation; these are influences which need to be considered in future experimental research.

## 5. Conclusions

This paper designs a pneumatic hazelnut harvester for mountains areas. In the experiments, the suspension velocity of hazelnuts was obtained as 15.6 $\mathrm{m \cdot s^{-1}}$ and the critical value of the suspension velocity of leaves was 4.92 $\mathrm{m \cdot s^{-1}}$. A coupled CFD-DEM method was used to simulate the effect of a pneumatic hazelnut harvester on the net fruit rate of hazelnuts under different combinations of operating parameters. In addition, a Box–Behnken design experiment was conducted with the sieve plate angle, the distance of the sieve plate, and the air flow velocity as factors and the net fruit rate of hazelnuts as indicators to explore the influence. The order of the factors affecting the index was air flow velocity > the sieve plate angle > distance of the sieve plate. The parameter optimization module of Design-Expert software was used to obtain the optimal combination of parameters for the factors: the sieve plate angle was 55.7°, the air flow velocity was 14.1 $\mathrm{m \cdot s^{-1}}$, and the distance of the sieve plate was 33.2 mm. The research results of this paper could provide a new design idea for optimizing the picking structure and sorting mechanism of the hazelnut harvester.

**Author Contributions:** Conceptualization: D.R. and H.Y.; software: H.Y. and R.Z.; validation: Y.Z. and H.Y.; supervisory role: W.W.; writing—review and editing: D.R.; investigation: J.Z.; data curation: F.L. and J.L.; funding acquisition and supervision: W.W. All authors have read and agreed to the published version of the manuscript.

**Funding:** This research was mainly supported by the Liaoning Xingliao Talent Program for Science and Technology Innovation Leaders (XLYC2002009).

**Institutional Review Board Statement:** Not applicable.

**Informed Consent Statement:** Not applicable.

**Data Availability Statement:** The datasets generated during and/or analyzed during the current study are available from the corresponding author upon reasonable request.

**Conflicts of Interest:** The authors declare no conflict of interest.

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
