# Peer review of "Research and Experiments of Hazelnut Harvesting Machine Based on CFD-DEM Analysis"

_agriculture, doi:10.3390/agriculture12122115_

Round 1
Reviewer 1 Report
The manuscript conducted theoretical and experimental research pertain to hazelnut harvesting machine.
Follwoings are my comments and questions.
1)
In the section "3.4. Parameter optimization and comparison experiments",
author conducted some opitimization. But I cannot find optimization methodology. Could you explain optimization method in detail?
2) I cannot find nomenclature in this paper. So it is hard to tell what the symbol indicates. You should add nomenclature at the end of manuscript.
3) In table 3, how do you determine dynamic and static friction coefficient?
4) In Figure 8, font in color bar is too small to tell value. You should use larger font in color bar.
Reviewer 2 Report
Manuscript concern experimental study and CFD-DEM modeling of work of the hazelnut harvesting machine. The scope of the study is very large. The study provide the set of optimal values of operating parameters of the harvesting machine. The Box-Behnken design test was used to find optimal values of parameters used for modelling. Although all steps of experimental study, CFD-DEM modelling and the optimization process were performed their description needs major revision, correction and improvement following particular comments.
Particular comments:
- Eq (1) – How the constant 0.06 of Eq. (1) was determined?
- Please check dimensional compliance of Eq. (1) and provide dimensions of all components of the equation.
- Table 2. How the elastic properties were determined? What is the unit of the Young’s modulus? Is it really the same for hazelnut and leaf?
- Terrible English. Please check grammar, spelling and punctuation of the entire manuscript.
- Most of sentences are too long. The example: “This paper will take the picking device and sorting device of the harvester as the carrier, select the rotational speed of the fan, the angle of the sieve leaf plate and the spacing between the vibrating screen and the sieve leaf plate as the test factors, take the net fruit rate of hazelnut as the test index, and conduct a three-factor, three-level Box-Behnken design test based on Rocky, in order to obtain the optimal combination of working parameters and provide theoretical reference for the subsequent device trial production as well as for The optimal combination of working parameters was obtained in order to provide theoretical reference for the subsequent trial of the device and also to guide and verify the performance of the field trial.” Is this truly the single one sentence? Please verify it.
- Table 3 and 4. Punctuation of the headings.
- Fig. 10 and 11. Y- axis: Material quality (kg)? Do you mean the mass of the material?
- Lack of comparison of simulated values with experimental (Fig. 10 and 11) to confirm quality of performed modeling.
- Quality of Fig. 12 should be improved. Units are non-visible at all.
- Total lack of discussion and comparison of results with finding of other researchers for similar studies.
- Conclusions should be shortened and focused on the novelty of the research.
Round 2
Reviewer 2 Report
Authors corrected the manuscript following my remarks. Now it is much better. Threfore, I can recommend it for publication in Agriculture.